# Fecal Excretion and Whole-Body Retention of Macro and Micro Minerals in Atlantic Salmon Fed Torula Yeast Grown on Sugar Kelp Hydrolysate

**DOI:** 10.3390/ani11082409

**Published:** 2021-08-14

**Authors:** Jon Øvrum Hansen, Sandeep Sharma, Svein Jarle Horn, Vincent G. H. Eijsink, Margareth Øverland, Liv Torunn Mydland

**Affiliations:** 1Department of Animal and Aquaculture Sciences, Faculty of Biosciences, Norwegian University of Life Sciences (NMBU), P.O. Box 5003, NO-1432 Aas, Norway; margareth.overland@nmbu.no (M.Ø.); liv.mydland@nmbu.no (L.T.M.); 2Faculty of Chemistry, Biotechnology and Food Science, Norwegian University of Life Sciences (NMBU), P.O. Box 5003, NO-1432 Aas, Norway; sansh@biomar.com (S.S.); svein.horn@nmbu.no (S.J.H.); vincent.eijsink@nmbu.no (V.G.H.E.)

**Keywords:** *Cyberlindnera jadinii*, iodine, mineral bioavailability, *Saccharina latissima*, *Salmo salar*

## Abstract

**Simple Summary:**

Cultivation of seaweed for various purposes has gained more focus in Europe during the last decades. Our study demonstrates the potential of seaweed as a substrate for yeast production, uptake of seaweed minerals into the yeast, and the bioavailability of minerals from this yeast in Atlantic salmon. We show that several minerals, especially the microminerals that are normally supplemented to commercial salmon diets, can be provided by yeast produced on seaweed hydrolysate.

**Abstract:**

Yeast is a microbial feed ingredient that can be produced from non-food biomasses. Brown seaweed contains high levels of complex carbohydrates that are not digested to any extent by monogastric animals but can be used as carbon sources for yeast production. The objective of this study was to investigate how minerals originating from brown macroalgae (*Saccharina latissima)* are incorporated in *Cyberlindnera jadinii* yeast and to assess the bioavailability of these different minerals as well as their accumulation into different organs of Atlantic salmon. The yeast *C. jadinii* was produced on a seaweed hydrolysate mixed with a sugar-rich wood hydrolysate in a 9:1 volume ratio and fed to Atlantic salmon (*Salmo salar*) in two different experiments: a digestibility experiment with 30% dietary inclusion of yeast and a retention experiment with increasing inclusion of yeast (5, 10, and 20%). Seaweed minerals such as zinc (Zn), copper (Cu), iodine (I), manganese (Mn), and cobalt (Co) were incorporated to a high degree in the yeast. The apparent fecal excretion of minerals was similar in both experiments, in general, with low excretion of, I, bromine (Br), and arsenic (As) (ranging from 18.0% to 63.5%) and high excretion of iron (Fe), Cu, Mn, aluminum (Al), cadmium (Cd) and lead (Pb) (ranging from 56.9% to <100%), despite the different fish size and fecal sampling method. High levels of Cu, I, Br, and Co in the yeast resulted in a linear decrease (*p* < 0.05) in retention of these minerals in salmon fed increasing levels of yeast. Despite increasing amounts of these minerals in the feed, whole-body levels of Cu and Mn remained stable, whereas whole-body levels of Co, somewhat unexpectedly, decreased with increased dietary yeast inclusion. The Cd from the yeast had low bioavailability but was concentrated more in the kidney (0.038 mg kg^−1^) and liver (0.025 mg kg^−1^) than in muscle (0.0009 mg kg^−1^). The given Cd level in fish strengthens the indication that it is safe to feed salmon with up to 20% inclusion of seaweed yeast without exceeding the maximum limit for Cd of 0.05 mg kg^−1^ w.w. in fish meat. The level and retention (*p* < 0.05) of As were lower in the yeast compared to fishmeal. The high level of iodine in *S. latissima* (3900 mg kg^−1^) was partly transferred to the yeast, and salmon fed increasing levels of yeast displayed a linear increase in whole-body I content (*p* < 0.05). There is, however, a need for a growth experiment with larger fish to draw any firm conclusions regarding food safety. Overall, this study shows that yeast grown on hydrolyzed seaweed can be a suitable mineral source for Atlantic salmon, especially when diets are low in fishmeal.

## 1. Introduction

Aquaculture can play a major role in meeting the global demand for protein for the growing human population. This may be dependent on sustainable feed resources that do not compete with the global food supply. Microbial ingredients from yeast, bacteria, or microalgae have the potential to fulfill this criterion. There has been increased focus on using organic-rich waste streams, mainly from food-related industries, for microbial ingredients production since this could reduce environmental problems and increase the recovery of nutrients and, thus, the sustainability of the total production chain [1,2,3].

Yeast represents a potential ingredient in aquafeeds due to its high protein content (45–60%) with favorable levels of histidine, isoleucine, and threonine but a lower level of methionine compared to fishmeal [4]. However, the cost of growth media components can constitute more than 50% of the overall cost for fermentative production of microbial biomass [5]. Hence, less expensive feedstocks for the cultivation of yeast need to be explored.

*Cyberlindnera jadinii* (previously classified as *Candida utilis* or Torula yeast) is a single-celled, protein-rich yeast belonging to the *Saccharomycetes* class. It has a status of generally regarded as safe (GRAS) and can metabolize a wide range of organic substrates, and has been widely used as a fodder yeast [6]. Molasses is a cheap by-product from the sugar industry and have been used worldwide for the production of both bioethanol and for the cultivation of *C. jadinii* [7]. In other studies, the cultivation of *C. jadinii* was performed using wood hydrolysates and/or sulfite spent liquor [3,8]. In a recent study, cultivation of *C. jadinii* was carried out using an enzymatic hydrolysate of *Saccharina latissima* as a source of nutrients and an enzymatic spruce hydrolysate as the main source of sugars for the fermentation [9]. One of the main challenges of using these complex sources of carbon and nutrients to cultivate *C. jadinii* is the risk of incorporation of unwanted components from the media that could adversely affect the growth performance and the safety and nutritional value of the fish product. Heavy metals and iodine (I) are known to accumulate in growing yeast [10,11]. In addition, yeast cells or cell wall products can absorb heavy metals [12]. Thus, one needs to consider, heavy metals, such as cadmium (Cd) and arsenic (As), as well as I, derived from the seaweed fermentation media, could be assimilated by *C. jadinii,* and be potentially harmful to fish and consumers [13].

The aim of the present study was to investigate how minerals originating from the brown seaweed *S. latissima* were incorporated in *C. jadinii* yeast, to analyze the bioavailability of these minerals in yeast-containing salmon feeds, and to assess their accumulation in different organs of yeast-fed Atlantic salmon. This research will increase the knowledge of using alternative marine substrates for microbial ingredient production and a key factor for increased use of green carbons and alternative mineral supplements in salmon feed.

## 2. Materials and Methods

### 2.1. Cultivation of Yeast

The present paper describes two fish feeding experiments where the yeast *C. jadinii* (CJS, LYCC 7549; Lallemand Yeast Culture Collection) was added as a dietary ingredient. The yeast was cultivated on a medium composed of enzymatic hydrolysates of *S. latissima* and spruce wood (*Picea abies*) mixed at a volume ratio of 9:1. The concentrated spruce hydrolysate (around 300 g L^−1^ glucose) was produced by the company Borregaard (Sarpsborg, Norway) in their biorefinery demonstration plant [14]. Upon mixing, the glucose concentration of the final medium was 38 g L^−1^. The yeast was grown in this medium in a 30 L bioreactor using a fed-batch fermentation procedure as described before [9]. After the completed fermentation, the yeasts cells were heat-inactivated, centrifuged, suspended in water, washed 3 times, and spray-dried [9]. The mineral composition of the yeast used in the digestibility and the retention experiment is given in Table 1. The 2 batches of yeast were produced with the same procedure but with different batches of seaweed hydrolysate, ending with a slight difference in mineral composition.

### 2.2. Biological Experiment and Facilities

The two fish feed experiments were performed at the Norwegian University of Life Sciences, Ås, Norway, which is an experimental unit approved by the National Animal Research Authority, Norway (permit no. 174). The experimental procedures were performed in accordance with the institutional and national guidelines for the care and use of animals (the Norwegian Animal Welfare Act and the Norwegian Regulation on Animal Experimentation). In both experiments, each diet was fed to triplicate tanks, in excess of appetite, i.e., 120% of expected feed intake, to ensure maximum voluntary feed consumption. Uneaten feed was collected with a retch wire screen, according to Shomorin et al. [15]. Dissolved oxygen was measured throughout the experiments and was kept above 8.5 mg L^−1^. Prior to sampling, the fishes were anesthetized with 60 mg L^−1^ Trikainmesilat (Finquel^®^, Scan Aqua, Årnes, Norway) in small aerated tanks.

#### 2.2.1. Digestibility Experiment

The digestibility experiment was performed with 240 Atlantic salmon pre-smolts with an average weight of 65 g that was distributed into 6 fiberglass tanks (300 L) and is the same fish experiment presented by Sharma et al. [9]. The average water temperature was 13.9 °C, and the fish were fed the experimental diets for 48 days. Feces were carefully stripped from all 40 fishes in each tank from the posterior as described by Austreng [16]. The stripped feces were immediately weighed and stored at −20 °C prior to freeze-drying.

A fishmeal-based control diet including yttrium oxide as a digestibility marker was mixed in a ratio of 70:30 with CJS according to Bureau and Hua [17]. The diet composition is presented in Sharma et al. [9]. The mineral compositions of the 2 diets are provided in Table 1. The diets were produced using gelatin as a binder, which was mixed in cold water and heated up to 60 °C in a microwave oven before mixing with dry ingredients and fish oil. The mash was cooled down to room temperature before pelleting using a 3 mm die (P35A, Carasco, Italy). The diets were kept frozen at −20 °C until the start of the experiment.

#### 2.2.2. Retention Experiment

In total, 4 diets were evaluated in a retention experiment with Atlantic salmon, where 3 diets with increasing inclusion levels (5%, 10%, 20%) of CJS were tested together with a control diet (Table 2). In these diets, protein from fishmeal, soy protein concentrate, and wheat gluten were substituted with protein from yeast in a ratio related to their protein contribution in the control diet. In addition, the diets were formulated to have a similar ratio of digestible protein to digestible energy by adjusting the level of cellulose and potato starch. The protein digestibility of the yeasts was not available prior to feed production, so an estimated protein digestibility for the yeast was set to 90%. In total, 720 Atlantic salmon with an average weight of 5.4 g and age of 24 weeks post-hatching were distributed into 12 tanks. The 80-L tanks were receiving approximately 4 L recirculated freshwater per min, and oxygen and water temperature were measured daily. The feeding experiment lasted for 42 days with an average water temperature of 14.2 °C.

All fish were pooled and weighed at the start of the experiment and on days 21 and 42. On day 42, the end of the experiment, 20 fish per tank were randomly selected and frozen at −20 °C and were later sampled for kidney, liver, and feces from the distal intestine in semi-frozen conditions. In addition, a transverse section of the muscle between the cranial part of the dorsal fin and the cranial part of the anal fin was taken without skin. These samples were pooled within the tank and freeze-dried prior to analysis. In addition, 15 fish at the start and 5 fish per tank at the end of the experiment were sampled for whole-body composition. The gastrointestinal tract was dissected out and rinsed for content with deionized water and included in the pooled sample of fish, followed by freeze-drying. After freeze-drying, the fish was ground using a Braun Minipimer 3 (MR320, Melsungen, Germany).

### 2.3. Chemical Analyses

The ingredients and diets were analyzed for dry matter by drying to constant weight at 104 °C, crude protein using Kjeldahl nitrogen × 6.25, and ash by incineration at 550 °C (Commission Regulation (EC) No 152/2009). Crude lipid was analyzed using an Accelerated Solvent Extractor (ASE200, Dionex, CA, USA). Starch content was analyzed enzymatically based on the use of thermostable α-amylase and amylo-glucosidase [18].

The minerals were analyzed by inductively coupled plasma spectrometry with mass spectrometric detection (ICP-MS) (PerkinElmer, MA, USA). For chlorine (Cl), bromine (Br), and I analysis, the samples were digested with concentrated 25% (*w*/*w*) tetramethylammonium hydroxide, and for other analyses, samples were digested with 65% HNO_3_ in a high-performance microwave reactor (UltraClave, MLS Milestone, Sorisole, Italy) [19,20]. The mineral analyses were validated using certified reference materials; NCS DC73349, NCS ZC73013 (National Analysis Center for Iron and Steel, Beijing, China), CRM GBW07603 (National Research Centre for CRM, Beijing, China), BCR 422, DORM-3, DOLT-5 (The European Virtual Institute for Speciation Analysis, EVISA), and 1577b (National Institute of Standards and Technology, MD, USA).

### 2.4. Calculation and Statistical Analysis

To evaluate the effect of the diets on the performance of the fish, several parameters were estimated. These were:Apparent fecal excretion = (−100 * ((*a* − *b*) × *a*^−1^)) + 100(1)
*a* = nutrient in feed × (yttrium oxide in feed)^−1^.
*b* = nutrient in feces × (yttrium oxide in feces)^−1^.
Feed conversion ratio (FCR) = feed consumed × weight gain^−1^(2)
Specific growth rate (SGR) = 100 × [(ln (final mean body weight) − ln (initial mean body weight)) × day^−1^](3)
Apparent retention of minerals = 100 × ((amount of mineral deposited in fish) × (amount of mineral ingested by the fish) ^−1^). Calculations were based on total biomass increase and total feed intake in each tank(4)

All statistical analyses were performed either using linear and quadratic regressions or one-way ANOVA followed by Tukey HSD as a post-hoc test. Regressions were used to observe the effect of the increased level of dietary yeast on the rate of mineral excretion, retention, and tissue content in Atlantic salmon. The best models were selected based on R^2^, residual plot, and significance level. All parameters were based on the tank as a statistical unit (*n* = 3), and the fish performance analyses were conducted with the General Linear Models procedure in the SAS software package (SAS/STAT Version 9.4. SAS Institute, Cary, NC, USA). Differences were considered significant when *p* < 0.05.

## 3. Results

Table 1 shows the mineral compositions of the ingredients and diets used in the present study, including the mineral compositions of *S. latissima*, fishmeal, and the *C. jadinii* yeast grown on *S. latissima*. Compared to the fish meal, the yeast had a numerically higher content of copper (Cu), I, manganese (Mn), cobalt (Co), nickel (Ni), and chromium (Cr), and lower content of calcium (Ca), phosphorous (P), potassium (K), and As.

Both the digestibility and the retention experiment were performed as planned, and there was no mortality of fish during the experimental period. The fish in the digestibility trial grew from an average of 65 to 95 g during the experiment with an SGR of 0.8. For the retention experiment, the overall performance of the fish was suitable, with an SGR ranging from 3.23 to 3.41 (Table 3). Final weights and SGR of fish fed increased levels of yeast had a trend to follow a quadratic pattern with an optimum of around 5% yeast inclusion. The FCR increased linearly from 0.58 to 0.66 in salmon fed an increased level of yeast.

The apparent fecal mineral excretion from the same experiment is presented in Table 4. The level of I excretion was 23.4% and 23.5% for the control and CJS fed fish, respectively. The fecal excretion of As and Br was low with approximately 15% and 35% for the control and CJS fed fish, respectively. The excretion of iron (Fe), Cu, aluminum (Al), and Cd was generally high (<84.6%) for both dietary treatments. The apparent fecal excretion of the minerals was also analyzed in the retention experiment, showing similarities with the digestibility experiment, namely low excretion of magnesium (Mg) I, Br, and As (ranging from 18.0% to 63.5%) and high excretion of Fe, Cu, Mn, Al, Cd, and lead (Pb) (ranging from 56.9% to <100%) (Table 5).

In-depth analysis of the accumulation of macro minerals in the retention experiment showed that whole-body contents of Ca, P, Cl, and Mg increased linearly with increased levels of dietary CJS inclusion (Table 6). As for the micro minerals, I and Cd content increased linearly, whereas Co and As decreased linearly in the whole body of salmon fed increased yeast inclusion. The level of Fe, Br, and Al followed a quadratic pattern with no clear optimum. The level of Fe and Br was highest in fish fed the 20% yeast inclusion shown by the ANOVA. Studies at the organ level showed that the content of K, sodium (Na), and Mg in the kidney followed a quadratic pattern with R^2^ values ranging from 0.34 to 0.36 for the macro minerals (Table 7). For the micro minerals in the kidney, Zn, Cu, Se, As, and Cd followed a quadratic curve with an average minimum level for fish fed 5% yeast. The content of I increased linearly from 2.2 to 8.7 mg kg^−1,^ and zinc (Zn) increased linearly from 220 to 280 mg kg^−1^. In muscle, there was a significant increase in selenium (Se) and I and a decrease in Se and Co with increased dietary yeast inclusion (Table 8). Among the macro minerals in the liver, the content of Ca, P, sulfur (S) and Mg, all fitted a quadratic regression line with a minimum level between 5% and 10% yeast inclusion (Table 9). The level of Zn and I increased linearly while the content of Mn, Br, and Cd followed a quadratic line with a minimum of 5% yeast inclusion. Levels for all analyzed minerals were numerically higher in the liver and kidney compared to muscle, except for As.

The apparent whole-body retention values provided in Table 10 show in percentage how much of the ingested minerals ended up being stored in the fish body. Some of the minerals show high retention levels, sometimes even exceeding 100%, such as P and K. The apparent retention of K and Na both followed a quadratic line with a maximum of around 10% yeast inclusion. The retention of Cl and S decreased, whereas P and Mg increased linearly in salmon fed increased levels of yeast. Regarding the micro minerals, the retention of Cu, I, Co, Br, and As decreased linearly in salmon fed increased yeast levels. Both I, Co, Br, and As had significantly higher retention in the control-fed group compared to the yeast-fed groups, as shown by ANOVA. The retention of Al was generally low (2.79–9.45) and followed a quadratic regression line with a minimum of 5% yeast inclusion. The retention of Mn, Cd, and Pb was generally low (<11.6%) and independent of dietary treatment.

## 4. Discussion

The present study focused on the accumulation of minerals originating from seaweed hydrolysate into *C. jadinii* yeast, and further, the bioavailability of these minerals in Atlantic salmon fed this yeast. Sharma et al. [9] showed that relatively high levels of I, Cr, As, Cu, Cd, and Br occur in hydrolyzed seaweed and in the produced yeast post-fermentation. This is in agreement with the known ability of live yeast cells to take up and accumulate a wide range of minerals, especially divalent cations such as Cu^2+^, Co^2+^, and Cd^2+^ [10,21]. Norris and Kelly [22] demonstrated for *Saccharomyces cerevisiae* that uptake of divalent cations was divided into two processes: first, a metabolism-independent accumulation (cation binding to molecules on the cell surface), followed by progressive, metabolism-dependent uptake. Failla et al. [23] showed that *C. jadinii* has a similar ability to accumulate cations as *S. cerevisiae*.

This ability to accumulate and absorb minerals has gained increased focus during the last years, with a focus on the removal of heavy metals from industrial wastewater [24]. The yeasts’ capability to take up minerals is also used to produce organic bound selenium yeast, where sodium selenite is added to the yeast growth medium, and Se is incorporated into the protein as selenomethionine [25]. Other minerals such as Zn, Mg, Cr, Co, I, Cr, and Mo could also be incorporated in yeast [26], and the resulting organically complexed minerals have shown promising bioavailability in chickens [27], rats [28], and for selenium in fish [29].

The protein digestibility of the yeast grown on seaweed and wood hydrolysates was 40.5%, indicating a low availability of the intracellular yeast proteins [9]. The low digestibility of unprocessed yeast can be explained by the intact robust cell walls of the yeast, which limit protein digestion [30,31]. While the bioavailability of minerals bound in the yeast may also be affected by the rigid cell wall, the present study showed that different minerals from the yeast have different bioavailability. When evaluating the performance parameters in the retention experiment, the lower protein digestibility of the yeast is reflected by the increased FCR in fish fed increased levels of yeast.

The level of macro minerals was numerically higher in the FM compared to the yeasts, except for S and Mg. Despite the lower levels of Ca and P in the yeast, there was increased retention and whole-body composition of Ca, P, and Mg in fish fed increasing levels of yeast. This was partly supported by the excretion values, apart from the high excretion levels in fish fed the highest yeast level. It is also worth mentioning that fecal excretion of minerals in fish can be confounded by the ability of fish to use additional minerals from the rearing water. Fecal excretion of Na was higher than 100%, implying that excretion of Na in the feces was greater than the level supplied through the diets. Therefore, the excess minerals in the feces might come from gill and skin uptake, which was not accounted for in the digestibility calculations.

Micro minerals such as Fe, Zn, Cu, Se, Mn, and Co are all essential for Atlantic salmon [32]. Recent research has shown that supplementation of micronutrients needs to be increased when diets high in plant proteins are used during the full production cycle of Atlantic salmon [33]. This is particularly relevant for minerals such as Zn and Se, which can be prevented taken up by phytic acid [34]. In the present experiment, Zn was effectively stored in the yeast, and yeast can thus be a suitable source of Zn in high plant diets. Fecal excretion of Zn varied a lot between the two experiments described above, with very high excretion levels in the retention trial. The reason for this is unclear but could be partly due to the different sizes (age) of fish. However, the level of Zn in the liver (on average 106 mg kg^−1^), kidney (on average 225 mg kg^−1^), muscle (on average 25 mg kg^−1^), and whole body (on average 145 mg kg^−1^) indicate that some Zn is absorbed and then excreted. Thus, the Zn bioavailability in the retention experiment is not as low as indicated by the fecal excretion values. The level of Mn and Co was high in the yeast compared to the FM. The dietary level of Co in the retention experiment was above the upper limit for fish feed, which is set at 1 mg kg^−1^. This may be caused by a possible high Co level in the mineral premix used since the screening of mineral premixes used in Norwegian salmon farming showed variation from 2 to 41 mg kg^−1^ [35]. The bioavailability of Co from this yeast was low, as indicated by a linear increase in fecal excretion and a decrease in retention when feeding increased levels of yeast.

The seaweed used in this study contained 1.1 mg Cd kg^−1^ DM, which is in line with the average content of 148 samples of *S. latissima* (0.94 mg kg^−1^) presented by Duinker et al. [36]. The level of Cd (0.78 mg kg^−1^) found in the final dried yeast shows that Cd, as a part of the divalent cations, had been taken up and stored by the live yeast as described by Brady et al. [10]. The Cd had low availability for the salmon in the present study, as shown by high fecal excretion and reduced retention with increased yeast inclusion. Despite the low availability, there was a small but significant increased level of Cd in the whole body. Especially the kidney had an increased level of Cd with increased yeast inclusion. This is in line with previous findings describing higher levels of Cd accumulation in the kidney and liver compared to muscle and the whole body of fish fed increased dietary levels of Cd [37,38]. In the present study, the diet with the highest yeast inclusion had a Cd level of 0.3 mg kg^−1^, which is low, but did lead to a detectable increase in the Cd content of the fish. In stark contrast to these observations, Berntssen et al. [38], when feeding Atlantic salmon with increasing levels of Cd (up to 250 mg kg^−1^ feed) for 4 months, did not observe significant accumulation in gut, kidney, nor muscle for levels below 5 mg Cd kg^−1^, which is a considerably higher than the levels used in the present experiment. This, together with the low absolute values of Cd in salmon muscle, strengthen the indication that it is safe to feed salmon with up to 20% inclusion of yeast grown on *S. latissima* hydrolysate without exceeding the maximum limit for Cd of 0.05 mg kg^−1^ w.w. in fish meat (EC no. 1881/2006).

The level of Pb and Al was also higher in the final yeast compared to FM, and the average level of Pb in *S. latissima* was found to be 0.33 mg kg^−1^ [36], which is twice the amount of what was found in the present yeast. Importantly, the levels of both Pb and Al were generally low, and the availability in the salmon was also low, as shown by high fecal excretion and low retention. The higher Pb level found in the kidney, compared to liver and muscle is in line with a study by Alves et al. [39], who showed that increased dietary Pb levels resulted in increased Pb accumulation in the kidney and liver of rainbow trout. The level of Pb in the muscle of fish fed the highest yeast diet was 0.0021 mg kg^−1^ w.w. and the maximum level allowed in fish muscle is 0.3 mg kg^−1^ w.w. (EC no. 1881/2006). It should, however, be specified that the average body weight of the present fish was only 22.5 g at the end of the trial and that a proper growth experiment with larger fish is needed to draw any firm conclusions regarding food safety.

The level of As can be high in *S. latissima,* ranging from 23 to 60 mg kg^−1^ in cultivated *S. latissima* and up to 95 mg kg^−1^ in the wild seaweed [40,41]. A major part of this As exists as organic forms, and thus, the level of the more toxic inorganic As is low in *S. latissima* [40,41]. In the present experiment, the level of total As was higher in the FM compared to the produced yeast with 11 and 4.2 mg kg^−1^, respectively. The level of As in FM was high but within the normal range found in Norwegian high-quality FM [42]. Both of the present experiments showed increased fecal excretion of As in yeast-fed fish compared to FM-fed fish. This indicates that the As in yeast has lower bioavailability compared to As from FM. This is supported by the observed decreased levels of As in both muscle and whole body. In comparison to other minerals, the relative abundance of As was higher in muscle compared to liver and kidney. The latter is in line with Francesconi et al. [43], who found increased levels of As in the muscle of yelloweye mullet after feeding with arsenobetaine, which is the most common form of As found in marine fish.

As expected, the level of I was high in the *S. latissima* (Table 1), and this led to the presence of 290 mg kg^−1^ in the yeast. Similarly, high levels of I were found in black soldier fly larvae that had been fed with the brown seaweed *Ascophyllum nodosum* [44]. In the retention experiment, the I content in the fish feed increased from 6.1 mg kg^−1^ in the control feed to 63 mg kg^−1^ for the diet with 20% yeast inclusion. This was reflected in a linear increased I content in all tested tissues, and the whole body of the Atlantic salmon fed increased yeast inclusion levels. The percentage of retained I decreased linearly with increased I intake, which indicates low bioavailability. The I level in both liver and kidney was close to five times higher compared to muscle and whole body. This is in line with Julshamn et al. [45], who described lower incorporation of I in muscle than in liver and kidney of Atlantic salmon that had been raised for 150 days in seawater and fed 54 or 86 mg I kg^−1^ diet. In accordance with the present results, experiments with rainbow trout in freshwater have shown increased I levels in the fish after feeding different seaweeds and that the I concentration in muscle was dependent on the dietary level of I and the length/percent body weight increase obtained [46,47].

A moderate I deficiency is present in many countries, especially for pregnant women [48]. Due to this, there has been a focus on enriching meat or fish with I, for which seaweed could be a suitable source. Julshamn et al. [45] showed that salmon fed an I level of 86 mg kg^−1^ obtained an I level in the wet fillet of 0.9 mg kg^−1^. A 200 g meal of this fillet would thus fulfill the recommended daily intake of 150 µg I [49]. In comparison, the highest I level in muscle in the present paper was 0.7 mg kg^−1^ w.w. and a 200 g fillet would thus almost fulfill the recommended daily intake of I. Of note, I levels in several wild marine fish, such as cod or pollack, can have much higher I levels, ranging from 3 to 22 mg kg^−1^ w.w. [50]. It is also worth noting that 40% of the commercial salmon feeds tested in 2019 had lower I contents (0.6–1.1 mg kg^−1^) than what is recommended for Chinook salmon reared in freshwater [35]. The latter underpins that yeast fermented on seaweed hydrolysate can be a great source of I supplementation to compensate for reduced levels of marine ingredients. If the I levels should become too high, pretreatment of seaweed, such as blanching in hot water [41], can effectively reduce these levels.

## 5. Conclusions

In conclusion, micro minerals such as Zn, Cu, I, Mn, and Co from the seaweed hydrolysate were incorporated to a high degree in the yeast. The apparent fecal excretion of minerals was similar in both experiments with Atlantic salmon, despite different fish sizes and fecal sampling methods. The Cd and As from the yeast had low bioavailability. The Cd was up-concentrated more in kidney and liver than in muscle, whereas it was the opposite for As. The concentration of I was high in both seaweed and yeast, and the I was further deposited in all organs of the salmon fed increased dietary I level. Overall, this study shows that yeast grown on hydrolyzed seaweed can be a suitable mineral source for Atlantic salmon.

## Figures and Tables

**Table 1 animals-11-02409-t001:** Macro and micro minerals present in the native *Saccharina latissima, Cyberlindnera jadinii* (CJS, produced from *S. latissima* and woody hydrolysates) used in either the digestibility (CJS_Dig_) or the retention experiment (CJS_Ret_) with Atlantic salmon (*Salmo salar*) and the fishmeal used in both experiments. Mineral composition of diets used in either the digestibility (Cont and CJS_30_) or the retention experiment with Atlantic salmon fed increasing level of CJS.

					Digestibility		Retention			
	Native *S. latissima*	FM	CJS_Dig_	CJS_Ret_	Cont ^a^	CJS_30_	Cont	5% CJS	10% CJS	20% CJS
Macro minerals		g kg^−1^								
Ca	48	35	2.4	5.7	20	17	18	17	16	17
P	4.4	22	3.7	1.0	12	9.8	12	11	11	11
K	96	13	7.8	9.1	6.8	7.7	7.4	6.9	7.1	8.1
Na	49	11	7.2	8.3	6	5.4	6.3	6.1	6	6.3
Cl	130	18	11	16.0	10	9.1	10	9.7	9.7	9.7
S	9.9	8.7	8.6	11.0	5.8	8.2	6	6.7	7.1	8.4
Mg	7.0	1.9	1.1	1.9	1.3	1.1	1.4	1.4	1.4	1.3
Micro minerals		mg kg^−1^								
Fe	120	100	190	140	120	100	120	150	160	170
Zn	46	61	120	72	210	150	210	200	200	210
Cu	2.4	2.6	13	16	12	11	11	11	12	14
Se	2.1	2.3	0.64	0.58	1.2	0.96	1.4	1.3	1.3	1.3
I	3900	2.5	410	290	6.1	88	6.1	21	35	63
Mn	4.7	3.2	7.7	53	26	30	31	32	34	38
Co	0.12	0.04	0.54	0.35	1.2	0.81	1.4	1.3	1.3	1.3
Ni	1.0	0.57	2.6	2.1	1.2	1.2	0.53	0.66	0.74	0.93
Br	1600	82	120	105	38	53	39	42	46	52
Cr	1.9	0.94	3.3	2.6	2.1	1.9	1.1	0.95	1.0	1.3
Al	68	3.8	3.1	8.9	67	37	76	91	81	83
As	44	11	3.8	4.2	6.3	5.2	5.7	4.6	4.5	4.7
Cd	1.1	0.18	0.34	0.78	0.10	0.31	0.13	0.18	0.23	0.3
Pb	0.40	0.03	0.039	0.16	0.28	0.16	0.16	0.24	0.23	0.22

^a^ Diet composition; fishmeal, 481.8 g kg^−1^; wheat gluten, 130.0 g kg^−1^; gelatinized potato starch, 120.0 g kg^−1^; fish oil, 150.0 g kg^−1^; vitamin and mineral premix, 70.0 g kg^−1^; MCP, 0.2 g kg^−1^; choline chloride, 2.0 g kg^−1^; yttrium oxide (Y_2_O_3_), 1.5 × 10^−3^ g kg^−1^. Detailed information on these ingredients can be found under Table 2.

**Table 2 animals-11-02409-t002:** Diet formulation and calculated chemical composition of the diets used in the retention experiment with Atlantic salmon (*Salmo salar*): a fishmeal control (FM), and three diets with increasing levels of *Cyberlindnera jadinii* (CJS).

Ingredients, g kg^−1^	Control	5% CJS	10% CJS	20% CJS
Fishmeal ^a^	450	432	414	379
CJS	0	50	100	200
Soy protein concentrate ^b^	50	48	46	42
Wheat gluten ^c^	90	86	83	76
Potato starch ^d^	120	104	87	63
Cellulose ^e^	50	40	30	
Gelatin ^f^	100	100	100	100
Fish oil ^g^	130	130	130	130
MCP ^h^	2	2	2	2
Choline ^i^	1.5	1.5	1.5	1.5
Mineral and vitamin premix ^j^	6.5	6.5	6.5	6.5
Yttrium oxide ^k^	0.15	0.15	0.15	0.15
Composition, g kg^−1^				
Dry matter	93.0	93.2	93.3	94.2
Crude protein	493	493	494	495
Crude lipid	177	176	175	173
Starch	103	92	79	64
Ash	22	26	30	37
DP: DE ratio ^l^	22.6	22.7	22.7	22.7

^a^ LT fishmeal, Norsildmel, Egersund, Norway; ^b^ soybean protein concentrate, Lyckeby Culinar, Fjälkinge, Sweden; ^c^ wheat gluten, Amilina AB, Panevezys, Lithuania; ^d^ Lygel F 60, Lyckeby Culinar, Fjälkinge, Sweden; ^e^ Alpha-Cel™ C100, International Fibre Europe NV, Temse, Belgium; ^f^ Rousselot^®^ 250 PS, Rousselot SAS, Courbevoie, France; ^g^ NorSalmOil, Norsildmel, Egersund, Norway; ^h^ monocalsium phosphate, Bolifor^®^ MCP-F, Oslo, Norway Yara; ^i^ choline chloride, 70% vegetable, Indukern s.a., Barcelona, Spain; ^j^ premix fish, Norsk Mineralnæring AS, Hønefoss, Norway. Per kg feed; retinol 3150.0 IU, cholecalciferol 1890.0 IU, α-tocopherol SD 250 mg, menadione 12.6 mg, thiamin 18.9 mg, riboflavin 31.5 mg, d-Ca-Pantothenate 37.8 mg, niacin 94.5 mg, biotin 0.315 mg, cyanocobalamin 0.025 mg, folic acid 6.3 mg, pyridoxine 37.8 mg, ascorbate monophosphate 157.5 g, Cu: CuSulfate 5H_2_O 6.3 mg, Zn: ZnSulfate 151.2 mg, Mn: Mn(II)Sulfate 18.9 mg, I: K-Iodide 3.78 mg, Ca 1.4 g; ^k^ yttrium oxide, Metal Rare Earth Limited, Shenzhen, China; ^l^ DP:DE = digestible protein: digestible energy ratio. Calculated based on internal values.

**Table 3 animals-11-02409-t003:** Weights, feed conversion ratio (FCR), and specific growth rate (SGR) for Atlantic salmon (*Salmo salar*) fed increasing dietary levels of *Cyberlindnera jadinii* (CJS) in the retention experiment.

	Control	5% CJS	10% CJS	20% CJS	SEM ^1^	*p*-Value ^2^	P	R^2^	P	R^2^
						ANOVA	Linear		Quadratic	
Start weight (g fish^−1^) 0 d	5.4	5.4	5.5	5.4	0.07	0.22	0.57	0.03	0.91	<0.01
Final weight (g fish^−1^) 42 d	22.5	22.7	22.8	21.2	1.1	0.38	0.14	0.20	0.084	0.27
Feed intake (g fish^−1^)	9.65	10.3	10.2	10.1	0.64	0.60	0.58	0.03	0.83	<0.01
FCR 0–42 d	0.58 ^a^	0.6 ^ab^	0.6 ^ab^	0.66 ^b^	0.02	0.0084	0.0007	0.70	0.0014	0.66
SGR 0–42 d	3.40	3.41	3.37	3.23	0.12	0.40	0.098	0.25	0.072	0.29

^1^ Pooled standard error of the mean; ^2^ *p*-value is given for ANOVA. Different letters indicate significant (*p* < 0.05) differences among diets within a row. *p*-value and R^2^ are given for linear and quadratic regression. *n* = 3 replicate tanks per treatment.

**Table 4 animals-11-02409-t004:** Apparent fecal excretion (% of ingested) of minerals from the digestibility experiment with Atlantic salmon (*Salmo salar*) fed 30% *Cyberlindnera jadinii* (CJS, produced from *S. latissima* and woody hydrolysates).

	Control	CJS30%	SEM ^1^	*p*-Value
Macro minerals	%			
Ca	95.9 ^a^	112 ^b^	2.89	0.0021
P	54.9 ^a^	51.3 ^b^	0.71	0.0036
K	4.8 ^b^	7.1 ^a^	0.33	0.0012
Na	104 ^b^	194 ^a^	6.3	0.0003
Cl	23.1 ^b^	42.0 ^a^	2.6	0.0009
S	32.9 ^b^	59.8 ^a^	1.56	<0.0001
Mg	33.7 ^b^	43.1 ^a^	1.56	0.0018
Micro minerals	%			
Fe	92.2	104	8.45	0.17
Zn	65.0 ^a^	55.5 ^b^	1.34	0.001
Cu	84.6 ^b^	112 ^a^	3.22	0.0005
Se	44.0 ^b^	58.2 ^a^	1.63	0.0004
I	23.4	23.5	0.56	0.89
Mn	88.7 ^b^	108 ^a^	2.3	0.0005
Co	81.8 ^b^	106 ^a^	3.9	0.0015
Ni	52.1 ^b^	105 ^a^	7.1	0.0008
Br	17.3 ^a^	34.8 ^b^	2.4	0.0008
Cr	51.4 ^b^	85.4 ^a^	7.2	0.0045
Al	108	120	22.0	0.52
As	12.8 ^b^	34.7 ^a^	0.59	<0.0001
Cd	97.4	92.5	2.3	0.059
Pb	68.3 ^b^	116.7 ^a^	5.7	0.0005

^1^ Pooled standard error of the mean. *p*-value given for the ANOVA where the different superscript indicates significant difference among diets (*p* < 0.05). *n* = 3 replicate tanks per treatment.

**Table 5 animals-11-02409-t005:** Fecal excretion (% of ingested) of macro and micro minerals from Atlantic salmon (*Salmo salar*) in the retention experiment where salmon were fed increasing levels of *Cyberlindnera jadinii* (CJS).

	Control	5% CJS	10% CJS	20% CJS	SEM ^1^	*p*-Value ^2^	P	R^2^	P	R^2^
Macro elements, g kg^−1^						ANOVA	Linear		Quadratic	
Ca	99.9	70.7	70.5	85.9	10.6	0.037	0.56	0.03	0.93	<0.01
P	82.5	67.9	65.5	90.0	8.9	0.038	0.33	0.10	0.10	0.25
K	76.7 ^a^	99.3 ^ab^	106 ^ab^	138 ^b^	17.1	0.016	0.0008	0.69	0.002	0.62
Na	111 ^a^	124 ^a^	128 ^ab^	145 ^b^	6.3	0.002	<0.0001	0.81	0.0005	0.72
Cl	16.3	16.6	17.1	18.8	1.88	0.50	0.11	0.23	0.1	0.24
S	48.9	58.7	66.3	75.8	10.8	0.073	0.0059	0.55	0.016	0.45
Mg	61.9 ^ab^	46.8 ^a^	47.3 ^a^	77.6 ^b^	10.4	0.029	0.12	0.22	0.027	0.40
Micro elements, mg kg^−1^										
Fe	120	83.6	91.8	94.0	14.5	0.11	0.28	0.11	0.56	0.03
Zn	132	138	146	202	40.8	0.225	0.038	0.36	0.027	0.40
Cu	65.8	103	108	102	20.6	0.11	0.12	0.21	0.34	0.09
Se	71.0	105	114	123	20.6	0.080	0.022	0.42	0.076	0.28
I	22.1	18.0	24.6	22.9	5.8	0.59	0.03	0.6	0.60	0.03
Mn	95.0	76.6	78.3	91.4	10.1	0.18	0.94	<0.01	0.53	0.04
Co	59.0 ^a^	88.1 ^ab^	100 ^b^	94.8 ^ab^	14.2	0.026	0.04	0.36	0.17	0.17
Ni	104 ^ab^	84.5 ^a^	119 ^b^	104 ^ab^	11.4	0.027	0.52	0.04	0.61	0.02
Br	20.3 ^a^	27.7 ^a^	38.1 ^ab^	52.9 ^b^	7.1	0.0034	<0.0001	0.79	<0.0001	0.80
Cr	99.4	81.0	130	99.2	24.3	0.23	0.70	0.01	0.87	<0.01
Al	109 ^b^	56.9 ^a^	84.9 ^ab^	94.0 ^ab^	16.9	0.037	0.65	0.02	0.65	0.02
As	29.9 ^a^	34.8 ^a^	40.6 ^ab^	63.5 ^b^	8.2	0.0071	0.0004	0.73	0.0002	0.75
Cd	149	242	236	239	53.5	0.21	0.15	0.19	0.34	0.09
Pb	109 ^b^	82.2 ^a^	85.3 ^a^	98.2 ^ab^	6.36	0.0046	0.65	0.02	0.79	<0.01

^1^ Pooled standard error of the mean; ^2^ *p*-value is given for ANOVA. Different letters indicate significant (*p* < 0.05) differences among diets within a row. *p*-value and R^2^ are given for linear and quadratic regression. *n* = 3 replicate tanks per treatment.

**Table 6 animals-11-02409-t006:** Macro and micro minerals present in the whole body of Atlantic salmon (*Salmo salar*) from the retention experiment where the salmon were fed increasing dietary level of *Cyberlindnera jadinii* (CJS).

	Control	5% CJS	10% CJS	20% CJS	SEM ^1^	*p*-Value ^2^	P	R^2^	P	R^2^
Macro elements ^3^, g kg^−1^						ANOVA	Linear		Quadratic	
Ca	8.8 ^a^	10.5 ^b^	11.2 ^b^	11.5 ^b^	0.74	0.0013	0.0016	0.65	0.021	0.42
P	11.7 ^a^	12.7 ^b^	13.3 ^bc^	13.7 ^c^	0.47	0.001	0.0004	0.73	0.008	0.52
K	11.8	12.3	11.8	12.7	0.43	0.072	0.071	0.29	0.054	0.32
Na	2.5	2.5	2.5	2.5	0.06	0.97	0.9	>0.01	0.83	<0.01
Cl	3.4 ^a^	3.2 ^ab^	3.1 ^ab^	3.1 ^b^	0.11	0.031	0.011	0.49	0.057	0.31
S	6.2	6.2	6.3	6.2	0.14	0.75	0.87	>0.01	0.95	<0.01
Mg	0.85 ^a^	0.89 ^ab^	0.91 ^bc^	0.94 ^c^	0.02	0.0026	0.0002	0.77	0.002	0.62
Micro elements, mg kg^−1^										
Fe	22.7 ^a^	24.0 ^a^	23.5 ^a^	27.7 ^b^	1.4	0.0081	0.0017	0.64	0.0007	0.70
Zn	142	142	150	145	9.2	0.68	0.55	0.04	0.71	0.01
Cu	3.0	2.7	2.7	2.7	0.4	0.78	0.47	0.05	0.62	0.02
Se	0.86	0.84	0.86	0.85	0.02	0.49	0.51	0.04	0.61	0.02
I	0.96 ^a^	1.3 ^b^	1.5 ^c^	2.2 ^d^	0.07	<0.0001	<0.0001	0.98	<0.0001	0.92
Mn	3.2	3.1	3.4	3.1	0.4	0.82	0.96	>0.01	0.94	<0.01
Co	0.23 ^a^	0.22 ^b^	0.19 ^c^	0.17 ^d^	0.008	<0.0001	<0.0001	0.91	0.003	0.74
Ni	0.075	0.074	0.092	0.099	0.02	0.56	0.16	0.18	0.19	0.16
Br	14.0 ^a^	14.0 ^a^	15.0 ^a^	17.0 ^b^	0.5	0.0001	<0.0001	0.82	<0.0001	0.91
Cr	0.11	0.15	0.14	0.15	0.04	0.77	0.4	0.06	0.56	0.03
Al	1.15	1.3	1.3	3.6	0.1	0.20	0.044	0.35	0.024	0.41
As	6.5 ^a^	5.4 ^b^	5.3 ^b^	5.0 ^b^	0.38	<0.0001	0.001	0.67	0.017	0.45
Cd	0.004 ^a^	0.007 ^bc^	0.005 ^ab^	0.009 ^c^	0.0015	0.0011	0.0029	0.61	0.0034	0.59
Pb	0.008	0.013	0.012	0.008	0.006	0.74	0.84	>0.01	0.63	0.02

^1^ Pooled standard error of the mean; ^2^ *p*-value is given for ANOVA. Different letters indicate significant (*p* < 0.05) differences among diets within a row. *p*-value and R^2^ are given for linear and quadratic regression. *n* = 3 replicate tanks per treatment; ^3^ Values are presented as pr kg freeze-dried material.

**Table 7 animals-11-02409-t007:** Macro and micro minerals present in the kidney of Atlantic salmon (*Salmo salar*) from the retention experiment where salmon were fed increasing dietary level of *Cyberlindnera jadinii* (CJS).

	Control	5% CJS	10% CJS	20% CJS	SEM ^1^	*p*-Value ^2^	P	R^2^	P	R^2^
Macro elements ^3^, g kg^−1^						ANOVA	Linear		Quadratic	
Ca	1.0	0.82	0.61	1.6	0.590	0.27	0.21	0.15	0.10	0.25
P	10.7	10.0	9.5	11.8	1.180	0.22	0.27	0.12	0.11	0.23
K	11.3	10.2	10.6	13.0	1.350	0.17	0.12	0.22	0.046	0.34
Na	3.7	3.1	3.6	2.7	0.400	0.054	0.048	0.33	0.039	0.36
Cl	4.4	3.7	4.6	4.0	0.450	0.10	0.8	<0.01	0.75	0.01
S	7.1	6.5	6.6	6.8	0.320	0.11	0.76	<0.01	0.82	<0.01
Mg	0.54	0.51	0.46	0.65	0.080	0.099	0.14	0.20	0.047	0.34
Micro elements, mg kg^−1^										
Fe	300	290	340	310	31	0.35	0.5	0.05	0.64	0.02
Zn	220	200	200	280	4.1	0.086	0.023	0.41	0.023	0.42
Cu	4.2 ^a^	3.7 ^a^	4.1 ^a^	5.6 ^b^	1.6	0.002	0.004	0.57	0.0003	0.74
Se	2.7	2.7	2.8	3.2	0.24	0.13	0.029	0.39	0.014	0.47
I	2.2 ^a^	3.9 ^b^	6.5 ^c^	8.7 ^d^	0.51	<0.0001	<0.0001	0.93	0.001	0.79
Mn	3.2	2.6	2.9	3.7	0.56	0.18	0.16	0.19	0.064	0.30
Co	0.77	0.63	0.74	0.62	0.07	0.079	0.14	0.21	0.17	0.18
Ni	0.064	0.096	0.034	0.058	0.005		0.16	0.23	0.14	0.26
Br	21	18	25	25	3.0	0.053	0.054	0.32	0.072	0.29
Cr	0.45	0.10	0.066	0.083	0.29	0.45	0.25	0.13	0.43	0.06
Al	0.0049	0.0007	0.0008	0.0011	0.003		0.32	0.11	0.33	0.10
As	2.7 ^ab^	2.4 ^ab^	2.3 ^a^	3.3 ^b^	0.32	0.042	0.085	0.27	0.019	0.44
Cd	0.023 ^a^	0.023 ^a^	0.025 ^ab^	0.038 ^b^	0.005	0.023	0.0032	0.60	0.0011	0.67
Pb	0.0081	0.0047	0.0063	0.013	0.0037		0.13	0.23	0.03	0.42

^1^ Pooled standard error of the mean; ^2^ *p*-value is given for ANOVA. Different letters indicate significant (*p* < 0.05) differences among diets within a row. *p*-value and R^2^ are given for linear and quadratic regression. *n* = 3 replicate tanks per treatment; ^3^ Values are presented as pr kg freeze-dried material.

**Table 8 animals-11-02409-t008:** Macro and micro minerals present in the muscle of Atlantic salmon (*Salmo salar*) from the retention experiment where salmon were fed increasing dietary level of *Cyberlindnera jadinii* (CJS).

	Control	5% CJS	10% CJS	20% CJS	SEM ^1^	*p*-Value ^2^	P	R^2^	P	R^2^
Macro elements ^3^, g kg^−1^						ANOVA	Linear		Quadratic	
Ca	0.56	0.82	0.56	0.45	0.19	0.28	0.28	0.11	0.20	0.16
P	11.3	12.0	11.3	11.7	0.52	0.36	0.79	<0.01	0.85	<0.01
K	20.0	20.3	19.7	20.7	0.93	0.65	0.49	<0.01	0.34	0.07
Na	1.9	1.8	1.8	1.8	0.1	0.27	0.15	0.19	0.33	0.09
Cl	2.4	2.2	2.1	2.2	0.14	0.12	0.072	0.19	0.21	0.15
S	7.4 ^ab^	7.6 ^a^	7.2 ^c^	7.5 ^ab^	0.21	0.053	0.98	<0.001	0.92	<0.01
Mg	1.1	1.2	1.2	1.2	0.06	0.16	0.18	0.17	0.23	0.14
Micro elements, mg kg^−1^										
Fe	7.2	6.9	8.9	7.0	1.7	0.50	0.96	<0.01	0.81	<0.01
Zn	26	26	24	23	1.7	0.30	0.057	0.32	0.085	0.27
Cu	1.4	1.3	1.4	1.4	0.1	0.56	0.44	0.06	0.28	0.11
Se	0.82 ^a^	0.80 ^ab^	0.73 ^b^	0.74 ^b^	0.026	0.018	0.01	0.49	0.048	0.33
I	0.70 ^a^	1.17 ^b^	1.63 ^c^	2.33 ^d^	0.13	<0.0001	<0.001	0.96	<0.001	0.84
Mn	0.68	0.73	0.68	0.63	0.067	0.49	0.27	0.12	0.19	0.16
Co	0.12 ^a^	0.11 ^ab^	0.10 ^bc^	0.083 ^c^	0.007	0.0007	<0.0001	0.83	0.001	0.67
Ni	n.d. ^4^	n.d.	n.d.	0.495						
Br	9.5	9.2	9.1	11.0	1.1	0.21	0.058	0.31	0.029	0.39
Cr	0.05 ^ab^	0.03 ^a^	0.03 ^a^	0.07 ^b^	0.01	0.006	0.11	0.23	0.017	0.44
Al	0.0006	n.d.	n.d.	0.0005						
As	10.3 ^a^	8.7 ^b^	8.2 ^b^	7.8 ^b^	0.44	0.0006	0.001	0.67	0.019	0.44
Cd	0.0005	0.0009	0.0008	0.0009	0.00026		0.6	0.04	0.78	0.01
Pb	0.007	0.003	n.d.	0.007			0.87	<0.01	0.78	0.03

^1^ Pooled standard error of the mean; ^2^ *p*-value is given for ANOVA. Different letters indicate significant (*p* < 0.05) differences among diets within a row. *p*-value and R^2^ are given for linear and quadratic regression. *n* = 3 replicate tanks per treatment; ^3^ Values are presented as pr kg freeze-dried material. ^4^ Not detected.

**Table 9 animals-11-02409-t009:** Macro and micro minerals present in the liver of Atlantic salmon (*Salmo salar*) from the retention experiment where salmon were fed increasing dietary level of *Cyberlindnera jadinii* (CJS).

	Control	5% CJS	10% CJS	20% CJS	SEM ^1^	*p*-Value ^2^	P	R^2^	P	R^2^
Macro elements ^3^, g kg^−1^						ANOVA	Linear		Quadratic	
Ca	0.43	0.34	0.28	1.13	0.38	0.086	0.045	0.34	0.013	0.47
P	11.7	11.7	11.7	12.7	0.58	0.16	0.041	0.35	0.020	0.43
K	12.3	12.0	12.7	12.7	0.5	0.36	0.23	0.14	0.24	0.13
Na	4.9	4.5	4.7	4.8	0.23	0.25	0.75	<0.01	0.9	<0.01
Cl	7.3	6.7	6.8	6.4	0.41	0.14	0.029	0.39	0.058	0.31
S	9.2 ^a^	8.6 ^b^	9.1 ^ab^	9.5 ^a^	0.22	0.006	0.15	0.37	0.048	0.33
Mg	0.6 ^ab^	0.6 ^ab^	0.5 ^a^	0.7 ^b^	0.038	0.028	0.23	0.14	0.065	0.30
Micro elements, mg kg^−1^										
Fe	140	150	160	177	0.038	0.69	0.20	0.16	0.22	0.14
Zn	102	98	91	133	17.1	0.065	0.014	0.45	0.017	0.45
Cu	33	30	31	32	5.3	0.9	0.97	<0.01	0.83	<0.01
Se	3.9	3.5	3.5	3.5	0.23	0.21	0.13	0.22	0.28	0.11
I	2.6 ^a^	4.9 ^b^	7.0 ^c^	10.3 ^d^	0.45	<0.0001	<0.0001	0.97	<0.0001	0.85
Mn	5.2 ^ab^	4.7 ^a^	5.3 ^ab^	7.1 ^b^	0.89	0.045	0.013	0.93	0.004	0.58
Co	0.54	0.48	0.45	0.50	0.078	0.53	0.67	<0.01	0.98	<0.01
Ni	0.037	0.049	n.d. ^4^	2.427			0.17	0.29	0.15	0.31
Br	31	30	33	40	0.005	0.17	0.028	0.39	0.018	0.44
Cr	0.049	0.029	0.050	0.093	0.043	0.38	0.13	0.21	0.084	0.27
Al	0.001	0.0004	0.0006	0.0052	0.004	0.059	0.094	0.28	0.055	0.35
As	2.6	2.5	2.5	2.7	0.18	0.54	0.41	0.06	0.25	0.13
Cd	0.0066	0.0061	0.0055	0.025	0.008	0.043	0.016	0.45	0.005	0.57
Pb	0.0036	0.0041	0.0036	0.0951	0.0042	0.74	0.15	0.24	0.12	0.26

^1^ Pooled standard error of the mean; ^2^ *p*-value is given for ANOVA. Different letters indicate significant (*p* < 0.05) differences among diets within a row. *p*-value and R^2^ are given for linear and quadratic regression. *n* = 3 replicate tanks per treatment; ^3^ Values are presented as pr kg freeze-dried material. ^4^ Not detected.

**Table 10 animals-11-02409-t010:** Retention of minerals (% of ingested) in Atlantic salmon (*Salmo salar*) from the retention experiment where salmon were fed increasing levels of *Cyberlindnera jadinii* (CJS).

	Control	5% CJS	10% CJS	20% CJS	SEM ^1^	*p*-Value ^2^	P	R^2^	P	R^2^
Macro elements, g kg^−1^						ANOVA	Linear		Quadratic	
Ca	35.3 ^a^	61.4 ^ab^	74.5 ^b^	64.7 ^b^	10.1	0.013	0.059	0.31	0.25	0.13
P	125 ^a^	153 ^ab^	167 ^b^	158 ^b^	10.7	0.011	0.044	0.35	0.21	0.15
K	261 ^ab^	283 ^a^	257 ^ab^	233 ^b^	16.5	0.061	0.044	0.35	0.019	0.43
Na	61.6 ^a^	60.8 ^ab^	61.8 ^a^	52.8 ^b^	3.1	0.031	0.019	0.44	0.003	0.59
Cl	56.6 ^a^	50.1 ^ab^	48.3 ^ab^	43.2 ^b^	2.8	0.0067	0.0005	0.72	0.057	0.32
S	162 ^a^	136 ^b^	133 ^b^	99.3 ^c^	6.8	<0.0001	<0.0001	0.89	0.0038	0.59
Mg	87.4	89.9	93.0	96.7	4.9	0.26	0.035	0.37	0.052	0.33
Micro elements, mg kg^−1^										
Fe	23.6	20.1	17.9	20.8	2.4	0.13	0.35	0.09	0.72	0.01
Zn	98.8	98.2	109	87.8	13.4	0.44	0.41	0.07	0.25	0.13
Cu	49.6	40.0	37.6	29.4	8.2	0.10	0.012	0.49	0.027	0.40
Se	109	106	110	99.3	5.3	0.21	0.096	0.25	0.059	0.31
I	31.2 ^a^	13.0 ^b^	9.36 ^c^	7.74 ^c^	1.8	<0.0001	0.0018	0.64	0.033	0.38
Mn	11.6	9.69	11.2	7.15	3.4	0.47	0.17	0.18	0.14	0.20
Co	43.2 ^a^	39.8 ^b^	35.9 ^c^	31.3 ^d^	1.4	<0.0001	<0.0001	0.94	<0.0001	0.81
Ni	26.9	20.0	24.2	19.8	8.6	0.76	0.46	0.05	0.52	0.04
Br	77.8 ^a^	68.0 ^b^	66.5 ^b^	65.6 ^b^	2.1	0.0008	0.006	0.55	0.055	0.32
Cr	17.1	27.6	25.8	19.3	11	0.67	0.97	<0.01	0.71	0.01
Al	2.9	2.76	3.17	9.54	3.8	0.21	0.053	0.33	0.029	0.39
As	222 ^a^	194 ^b^	197 ^b^	156 ^c^	8.4	0.0001	<0.0001	0.85	<0.0001	0.80
Cd	−0.27 ^a^	4.24 ^b^	1.22 ^ab^	4.05 ^b^	1.47	0.0071	0.083	0.27	0.14	0.20
Pb	9.58	11.0	10.2	6.56	6.2	0.87	0.49	0.05	0.42	0.07

^1^ Pooled standard error of the mean; ^2^ *p*-value is given for ANOVA. Different letters indicate significant (*p* < 0.05) differences among diets within a row. *p*-value and R^2^ are given for linear and quadratic regression. *n* = 3 replicate tanks per treatment.

## Data Availability

Data is available on request from the authors.

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
