# Peer review of "Fecal Excretion and Whole-Body Retention of Macro and Micro Minerals in Atlantic Salmon Fed Torula Yeast Grown on Sugar Kelp Hydrolysate"

_animals, 2021, doi:10.3390/ani11082409_

Round 1

Reviewer 1 Report

This is a companion paper to a previous study by many of the same authors published in 2018, which established the experimental diet used in the present study, and includes data gathered simultaneously to that work. The experimental design is sound and the results presented very logically. I have only a few comments for consideration by the authors, which are itemized below.

General Comments:

The substantial amount of data presented makes it difficult to ascertain the reasoning of why certain results have been chosen to be included in the Discussion, while others receive very little attention. It appears that the authors have selected which results to highlight primarily based on perceived relevance to the end goal of potential use of CJS as an alternative energy source in fish feed, and implications for animal growth and safety for consumption. I suggest adding some general discussion, especially for the macrominerals, at least providing relevant references to previous work for readers who would like better context for those results. In short, if the results are considered important enough to be included within a table, they should be mentioned and contextualized.

Similarly, there is little mention of the impact of the different CJS levels in the experimental diets of the retention study (Table 2). Can any conclusions be drawn with regard to recommended doses of this ingredient?

On page 4, adjust punctuation to give equations 1, 4, & 5 as equations. That is, with = rather than : and omitting ending stops.

Typos:

Line 17: change to “microminerals that are”

Line 36: delete “up”

Line 392: “reduce” rather than “reducing”

Author Response

This is a companion paper to a previous study by many of the same authors published in 2018, which established the experimental diet used in the present study, and includes data gathered simultaneously to that work. The experimental design is sound and the results presented very logically. I have only a few comments for consideration by the authors, which are itemized below.

General Comments:

The substantial amount of data presented makes it difficult to ascertain the reasoning of why certain results have been chosen to be included in the Discussion, while others receive very little attention. It appears that the authors have selected which results to highlight primarily based on perceived relevance to the end goal of potential use of CJS as an alternative energy source in fish feed, and implications for animal growth and safety for consumption. I suggest adding some general discussion, especially for the macrominerals, at least providing relevant references to previous work for readers who would like better context for those results. In short, if the results are considered important enough to be included within a table, they should be mentioned and contextualized.

The authors agree with the reviewer, and have added the following to the manuscript:

The level of macro minerals was numerically higher in the FM compared to the yeasts, except for S and Mg. Despite the lower levels of Ca and P in the yeast, there was an increased retention and whole-body composition of Ca, P, and Mg in fish fed increasing level of yeast. This was partly supported by the excretion values, apart from the high excretion levels in fish fed the highest yeast level. It is also worth mentioning, that fecal excretion of minerals in fish can be confounded by the ability of fish to utilize additional minerals from the rearing water. Fecal excretion of Na was higher than 100%, implying that excretion of Na in the feces was greater than the level supplied through the diets. Therefore, the excess minerals in the feces might come from gill and skin uptake, which was not accounted for in the digestibility calculations.

To our knowledge there are no published similar experiments, and the only article concerning yeast and digestibility of minerals in salmonids is only examining Se (https://onlinelibrary.wiley.com/doi/10.1111/j.1439-0396.2008.00888.x). However, we have internal unpublished data demonstrating different fecal excretion of Ca, P, and Mg in salmon fed different type of yeasts that will be published during the next year.

Similarly, there is little mention of the impact of the different CJS levels in the experimental diets of the retention study (Table 2). Can any conclusions be drawn with regard to recommended doses of this ingredient?

The authors partly agree, but this was designed as a regression experiment to look at the availability of different minerals in the present yeast. This trial was performed to test a proof of concept by feeding salmon yeast grown on seaweed hydrolysate. Thus, both the fermentation media and -protocol was not optimized. As the aim in the article is stating, the intension was to look at the incorporation of minerals from the seaweed to the yeast, and then, how they further were incorporated in the salmon. The fish was fairly small at the end of the trial and the processing of the yeast was not optimized which led to a low protein digestibility. Thus, any further recommendation of doses will be challenging at this stage. However, the level of the different minerals and their maximum limits in both feed and fish have been discussed, without mentioning exact yeast inclusion levels.  

On page 4, adjust punctuation to give equations 1, 4, & 5 as equations. That is, with = rather than : and omitting ending stops.

The authors agree and have made suggested corrections.

Typos:

Line 17: change to “microminerals that are”

Line 17: is changed to “microminerals that are”

Line 36: delete “up”

Line 36: “up” is deleted

Line 392: “reduce” rather than “reducing”

Line 392: “reducing” is changed to “reduce”

Reviewer 2 Report

The authors investigated fecal excretion and whole-body retention of macro and micro minerals in Atlantic salmon fed Torula yeast grown on sugar kelp hydrolysate. They designed four treatments to comprehensively test their hypothesis regarding the optimum dosage of yeat in the salmon diet to provide maximum mineral retention. This manuscript (MS) was clearly written and easy to understand. This work can help the sustainability of this species farming as using natural bioresources is required for the future of aquaculture. However, some major issues significantly compromised the quality of this MS.

Major comments:

  • This is a big question of whether enriching salmon with some minerals and heavy metals are safe for humans who consume this fish or not. There are some interesting options for you to complete the puzzle, such as reporting global and national standards of these heavy metals and see whether the concentrations in your study were in a healthy range or not. Then, you can use risk assessment of human health (Target Hazard Quotient) as well. I commented in the results section about it.
  • The format of tables should be updated, and the authors need to understand some terms around regression, such as linear and quadratic. They have a separate P-value and mentioning only polynomial relation is not enough. I added a reference in the results section for your ease.
  • Further, when it was a quadratic relation, the authors can suggest the optimum level of yeast in diet regarding that parameters. You can add this information in tables by adding a column.

However, I have touched on some more points that can contribute to the improvement of this MS.

Minor comments

Abstract

  • Line 17, change to “We show that several minerals”.
  • Line 23, please add brown algae before Saccarina latissima.
  • Line 27, change from “2” to “two.”
  • I suggest adding the complete name of minerals as well.
  • Please add P-value to the abstract, for example, lines 33, 32, 35, etc.
  • Line 35-36, please mention whether it was in a healthy range for humans in terms of bioaccumulation or not; please add it for other heavy metals as well.
  • Line 41, please add it the low fish meal bring in a mineral deficiency.
  • Please reorder the keywords alphabetically and capitalize each word.
  • Please write the abstract more numerically about the results. You can do it by adding their numbers in parentheses.

Introduction:

  • Well-developed section, clear fellow and relevant points were included.
  • Line 47-49, please revise it as this is not clear.
  • Line 48 and elsewhere, if you use any abbreviation less than three times in the MS, you can use the complete form instead of the abbreviation.
  • Line 54, please add protein contents and mention they are rich in which amino acids
  • Please mention the novelty of your work in the last paragraph of the introduction.
  •  

Material and methods

  • Well-organized section. Clear fellow and all required details were provided.
  • Line 113, please add the diet composition in the bottom or footnote of Table 1 as is required to be here; in this paper.
  • Please rewrite the formula; they are too messy.
  • Line 163, quadratic polynomial regressions instead of “2nd degree”?
  •  

Results

  • Well-written section, all necessary information have been covered but need to be more numeric.
  • Line 175, for the first time in the MS, please write the complete form of minerals along with abbreviations, and for the rest, you can use abbreviations.
  • Line 181, change “2nd polynomial” to quadratic.
  • Line 182, here and other parts of MS, the results are blind; which treatments? Was increased from where to where? Please add more specific information to the results section.
  • Line 183-185, please move it to the discussion.
  • 196, I think you meant the quadratic trend.
  • Line 198, polynomial pattern does not say anything. Please use the appropriate terms which are “linear relation” or “quadratic relation,” throughout the MS.
  • Line 199 and elsewhere, when you say “quadratic relation,” it means a number has been max or min. You can use “parabola opens upward” and “parabola opens downward” to give complete information to readers. Further, please mention in which treatment we had max and min. Please keep doing that for all mentioned parameters.
  • When you compared the data of Table 1 regarding fish meal and yeast, did you use the statistical analysis? If not, you can use the T-Test for your claim to say it was “significantly different”—the same scenario for comparing retention and digestibility data. If you did regression analysis for retentions, please add the P-value of ANOVA, Linear, and Quadratic relations. You can use this paper as a reference to see how you can report them. (https://www.sciencedirect.com/science/article/pii/S0044848619305861
  • Please change the format of tables like the mentioned paper as a standard way of reporting both ANOVA and regression (if you wanted to benefit from both approaches). The way you have reported is not correct.
  • Please highlight this important output in the discussion: Although there was no significant difference in growth and FCR, retention was different. However, it is a good sign that you could use yeast without any negative impacts on growth performance.
  • Table 4, what 194% or higher than 100% means for fecal excretion? Please explain somewhere in the MS.
  • Another thing which can be useful for this experiment is to calculate whether with feeding yeast rich of heavy metals, is any problem for human when we feed salmon?. There are some interesting options for you to complete the puzzle, such as reporting global and national standards of these heavy metals and see whether the content in your study was in a healthy range or not. Then use Risk assessment of human health (Target Hazard Quotient). You can find them in any heavy metal papers like this (https://www.sciencedirect.com/science/article/pii/S2352485520306800?via%3Dihub ), or you can find other parameters from other articles.

Discussion

  • I suggest putting the subheading for the discussion section like results, especially for trials. Also, keep a sequence in subheading for investigated factors, in material and method, result, and discussion.
    • Line 304-305, was significant? If not, please delete it.
    • Line 312, if you still have samples of diets, adding phytic acid contents of diet can help you a lot to see how they react with each other.
    • Please add a section as risk assessment of human health in the discussion.
    • I suggest adding a short sentence in starting any paragraph that which metal is good, and which one is bad. It will help readers to understand what is going on from a health point of view.
    • You can move some of the sentences related to health to “risk assessment of human health” section.
    • Line 390, please be consistent with I or iodine and also for other minerals.

Best regards

Author Response

The authors investigated fecal excretion and whole-body retention of macro and micro minerals in Atlantic salmon fed Torula yeast grown on sugar kelp hydrolysate. They designed four treatments to comprehensively test their hypothesis regarding the optimum dosage of yeat in the salmon diet to provide maximum mineral retention. This manuscript (MS) was clearly written and easy to understand. This work can help the sustainability of this species farming as using natural bioresources is required for the future of aquaculture. However, some major issues significantly compromised the quality of this MS.

Major comments:

  • This is a big question of whether enriching salmon with some minerals and heavy metals are safe for humans who consume this fish or not. There are some interesting options for you to complete the puzzle, such as reporting global and national standards of these heavy metals and see whether the concentrations in your study were in a healthy range or not. Then, you can use risk assessment of human health (Target Hazard Quotient) as well. I commented in the results section about it.

The authors agree highly with the reviewer that it is important to have focus on the human health safety with regards to the heavy metals. With this the authors have mentioned this when discussing the Cd and I levels in the experimental fish. In addition, the authors have added this for the Pb discussion:

   The level of Pb in muscle of fish fed the highest yeast diet was 0.0021 mg kg-1 w.w. and the maximum levels allowed in fish muscle is 0.3 mg kg-1 w.w. (EC No. 1881/2006). It should, however, be specified that the average body weight of the present fish was only 22.5 g at the end of the trial, and that a proper growth experiment with larger fish is needed to draw any firm conclusions regarding food safety.

The authors find it difficult to do a user risk assessment of human health (Target Hazard Quotient) when the fish has been fed from average 5.5 to 22.5 grams, which is far from a normal slaughter size of 5-6kg. This trial was performed to test a proof of concept by feeding salmon a yeast grown on seaweed hydrolysate. Thus, neither the fermentation media nor -protocol was optimized. As the aim in the article is stating, the intension was to look at the incorporation of minerals from the seaweed to the yeast, and then, how they further were incorporated in the salmon. The authors hope this is ok, and future work is needed to optimize this protocol and test it out in a longer growth experiment with larger salmon to perform such a risk assessment of human health (Target Hazard Quotient).  

  • The format of tables should be updated, and the authors need to understand some terms around regression, such as linear and quadratic. They have a separate P-value and mentioning only polynomial relation is not enough. I added a reference in the results section for your ease.

The authors agree and have done changes accordingly.

  • Further, when it was a quadratic relation, the authors can suggest the optimum level of yeast in diet regarding that parameters. You can add this information in tables by adding a column.

The authors agree and have done changes accordingly.

However, I have touched on some more points that can contribute to the improvement of this MS.

Minor comments

Abstract

  • Line 17, change to “We show that several minerals”.
  • Line 17, is changed to “We show that several minerals”.
  • Line 23, please add brown algae before Saccarina latissima.
  • Line 23 is changed accordingly (brown macroalgae)
  • Line 27, change from “2” to “two.”
  • Line 27 is changed accordingly
  • I suggest adding the complete name of minerals as well.
  • The abstract is changed accordingly
  • Please add P-value to the abstract, for example, lines 33, 32, 35, etc.
  • P-values have been added to the abstract
  • Line 35-36, please mention whether it was in a healthy range for humans in terms of bioaccumulation or not; please add it for other heavy metals as well.

Line 35-36, corrections have been added: “With the given Cd level in fish strengthen the indication that it is safe to feed salmon with up to 20% inclusion of seaweed yeast without exceeding the maximum limit for Cd of 0.05 mg kg-1 w.w. in fish meat.

In addition, this line have been added to the abstract: “There is, however, a need for a growth experiment with larger fish to draw any firm conclusions regarding food safety”

  • Line 41, please add it the low fish meal bring in a mineral deficiency.

The authors disagree and find this not to be the main topic of the manuscript and hope to let this be left out of the abstract.

  • Please reorder the keywords alphabetically and capitalize each word.

The keywords are corrected

  • Please write the abstract more numerically about the results. You can do it by adding their numbers in parentheses.

The authors agree and have corrected accordingly.

Introduction:

  • Well-developed section, clear fellow and relevant points were included.
  • Line 47-49, please revise it as this is not clear.

Line 47-49 is further revised.

  • Line 48 and elsewhere, if you use any abbreviation less than three times in the MS, you can use the complete form instead of the abbreviation.

The authors agree and have corrected abbreviations

  • Line 54, please add protein contents and mention they are rich in which amino acids

The authors agree and have done following corrections: “Yeast represents a potential ingredient in aquafeeds due to its high protein content (45-60%) with favorable levels of histidine, isoleucine and threonine, but lower level of methionine, compared to fishmeal

  • Please mention the novelty of your work in the last paragraph of the introduction.
  • This have been added to the last section;“This research will increase the knowledge of using alternative marine substrates for microbial ingredient production, and a key factor for increased use of green carbons and alternative mineral supplements in salmon feed.”

Material and methods

  • Well-organized section. Clear fellow and all required details were provided.
  • Line 113, please add the diet composition in the bottom or footnote of Table 1 as is required to be here; in this paper.

Line 113; diet composition is added in a footnote.

  • Please rewrite the formula; they are too messy.

The formulas are re-organized.

  • Line 163, quadratic polynomial regressions instead of “2nd degree”?

Line 163, this is renamed in the whole manuscript.

Results

  • Well-written section, all necessary information have been covered but need to be more numeric.
  • Line 175, for the first time in the MS, please write the complete form of minerals along with abbreviations, and for the rest, you can use abbreviations.

Line 175, the abbreviations of minerals is changed accordingly

  • Line 181, change “2nd polynomial” to quadratic.

Line 181; corrected

  • Line 182, here and other parts of MS, the results are blind; which treatments? Was increased from where to where? Please add more specific information to the results section.

Line 182, the authors have added “from 0.58 to 0.66”. and have also in general added more specific information throughout the result section. This required several changes throughout the results section and is only assigned using the “track changes” in the revised manuscript.

  • Line 183-185, please move it to the discussion.

The lines are removed, and the reference is moved to line 336 in the discussion.

  • 196, I think you meant the quadratic trend.

Line 196 is corrected.

  • Line 198, polynomial pattern does not say anything. Please use the appropriate terms which are “linear relation” or “quadratic relation,” throughout the MS.

Line 196; “quadratic relation” is used throughout the MS

  • Line 199 and elsewhere, when you say “quadratic relation,” it means a number has been max or min. You can use “parabola opens upward” and “parabola opens downward” to give complete information to readers. Further, please mention in which treatment we had max and min. Please keep doing that for all mentioned parameters.

Line 199; “quadratic relation” is used throughout the MS. Instead of using “parabola opens upward” and “parabola opens downward”, the authors have indicating if it is an “optimum” or “minimum” which describe the same information to the readers. We hope this is ok.

  • When you compared the data of Table 1 regarding fish meal and yeast, did you use the statistical analysis? If not, you can use the T-Test for your claim to say it was “significantly different”—the same scenario for comparing retention and digestibility data. If you did regression analysis for retentions, please add the P-value of ANOVA, Linear, and Quadratic relations. You can use this paper as a reference to see how you can report them. (https://www.sciencedirect.com/science/article/pii/S0044848619305861

When we were comparing the data of Table 1 regarding fish meal and yeast, we did not use statistical analysis since this is numbers from material that we don’t have any replicates. The fish meal is not produced by us and to use analyzed values from one type of fish meal as replicates is not appropriate.

The authors agree with the second comments and have adjusted all tables accordingly.

  • Please change the format of tables like the mentioned paper as a standard way of reporting both ANOVA and regression (if you wanted to benefit from both approaches). The way you have reported is not correct.

The authors agree and have adjusted all tables accordingly.

  • Please highlight this important output in the discussion: Although there was no significant difference in growth and FCR, retention was different. However, it is a good sign that you could use yeast without any negative impacts on growth performance.

The authors have described the low protein digestibility and the increase in FCR in fish fed increased yeast inclusion and mention the trend for an optimum in growth at 5% yeast inclusion. We hope this is enough and it is not totally correct to state that there was no negative impact on growth performance.

  • Table 4, what 194% or higher than 100% means for fecal excretion? Please explain somewhere in the MS.

Line 349-354: The authors agrees, and have added: ”It is also worth mentioning, that fecal excretion of minerals in fish can be confounded by the ability of fish to utilize additional minerals from the rearing water. Fecal excretion of Na was higher than 100%, implying that excretion of Na in the feces was greater than the level supplied through the diets. Therefore, the excess minerals in the feces might come from gill and skin uptake, which was not accounted for in the digestibility calculations.

  • Another thing which can be useful for this experiment is to calculate whether with feeding yeast rich of heavy metals, is any problem for human when we feed salmon?. There are some interesting options for you to complete the puzzle, such as reporting global and national standards of these heavy metals and see whether the content in your study was in a healthy range or not. Then use Risk assessment of human health (Target Hazard Quotient). You can find them in any heavy metal papers like this (https://www.sciencedirect.com/science/article/pii/S2352485520306800?via%3Dihub ), or you can find other parameters from other articles.

Please look at the feed back for the major comment in the start of the letter. 

Discussion

  • I suggest putting the subheading for the discussion section like results, especially for trials. Also, keep a sequence in subheading for investigated factors, in material and method, result, and discussion.

The authors hope it is ok to keep the structure as it is for det discussion. The digestibility experiment is fairly small compare to the retention experiment, and thus, a section of this in the discussion will not be optimal. Furthermore, the discussion is not very long, and the results of each experiment is good to discuss simultaneously. Thus, the authors hope, and suggests, to keep the discussion as it is.

  • Line 304-305, was significant? If not, please delete it.

Line 304-305, the result was significant, and we hope to keep it like this.

  • Line 312, if you still have samples of diets, adding phytic acid contents of diet can help you a lot to see how they react with each other.

Line 312, unfortunately, the phytic acid was not measured. Due to the lack of in-house analysis and the short deadline for re-submission, the authors suggest not to spend time on this since it is not a key point for the paper.

  • Please add a section as risk assessment of human health in the discussion.

Please see the main comment in the start of the letter

  • I suggest adding a short sentence in starting any paragraph that which metal is good, and which one is bad. It will help readers to understand what is going on from a health point of view.

The authors partly agree, but as we said in the start of the letter, the main focus for this paper was not human health. We have discussion and mention that the micro minerals like Fe, Zn, Cu, Se, Mn and Co are all essential for Atlantic salmon, and discussed the limits for heavy metals and I in fish and further how this dietary yeast inclusion could affect this. We hope this is ok, and that a more health question/problem should be investigated in a proper growth trial with bigger fish later in the future.

  • You can move some of the sentences related to health to “risk assessment of human health” section.
  • Line 390, please be consistent with I or iodine and also for other minerals.

Line 390, is changed accordingly.

Round 2

Reviewer 2 Report

The authors have improved the quality of the MS and I suggest authors reading one more time to fix few language errors. I can see some errors which are required to fix. Then, it would be ready for the final steps for acceptance.

  • Please change Anova to ANOVA throughout the MS.
  • Please delete the numbers from the Keyword
  • Regarding P-value, please be consistent with (P<0.05) throughout the text of the MS.
  • Line 206 “Iand Cd content increased linearly (P=<.0001 and 0.0029)”. For the cases like that, no need to report P-value. When you mention is linear relation, it already means it has been significant. Please update the MS from this point.
  • Line 262, change (P≤.05) to “P<0.05”

Author Response

The authors have improved the quality of the MS and I suggest authors reading one more time to fix few language errors. I can see some errors which are required to fix. Then, it would be ready for the final steps for acceptance.

  • Please change Anova to ANOVA throughout the MS.

The changes regarding “ANOVA” is done throughout the MS.

  • Please delete the numbers from the Keyword

The numbers are removed.

  • Regarding P-value, please be consistent with (P<0.05) throughout the text of the MS.

The P-value is corrected throughout the manuscript.

  • Line 206 “Iand Cd content increased linearly (P=<.0001 and 0.0029)”. For the cases like that, no need to report P-value. When you mention is linear relation, it already means it has been significant. Please update the MS from this point.

Line 206: the p-values are corrected

  • Line 262, change (P≤.05) to “P<0.05”,

Line 262, (P≤.05) is changed to “P<0.05”, and the same for all tables.